

# Pain and satisfaction: the case of isolated COVID-19 patients of Pakistan

Wajiha Haq[1]  Siddrah Irfan[2]  Syed Hassan Raza[3]  Waqar Ahmed[4]  Mian Alam[5]
Samia Wasif[6]  Muhammad Wasif Malik[7]  Saeed Ahmad[8]

[1] Department of Economics, School of Social Sciences and Humanities, National University of Sciences and Technology, Islamabad, Punjab, Pakistan
[2] Department of Behavioral Sciences, School of Social Sciences and Humanities, National University of Sciences and Technology, Islamabad, Pakistan
[3] School of Economics, Quaid-i-Azam University, Islamabad, Pakistan
[4] District Health Authority, Rawalpindi, Pakistan
[5] Corona Management Centre, Rawalpindi Institute of Urology & Transplantation, Rawalpindi, Pakistan
[6] Department of Humanities, COMSATS University, Islamabad, Pakistan
[7] Field Epidemiology & Disease Surveillance Division, National Institute of Health, Islamabad, Pakistan
[8] Public Health England, Islamabad, Pakistan

Corresponding author
Wajiha Haq,
dr.wajihahaq@s3h.nust.edu.pk

## ABSTRACT

**Background.** Over 3 million people lost their lives across the globe due to the COVID-19 related pandemic. The unprecedented restrictions on mobility have imposed in the form of isolation, quarantines, and lockdowns to curb the profound effects of the pandemic and retain physical health. This study examined the relationship between various pain levels, demographic variables and patient satisfaction with COVID-19 during isolation.

**Methods.** The primary data in this study were collected from isolated patients during their isolation and treatment in a public hospital. We obtained information from patients with COVID-19 (N = 100) about their demographic characteristics, varying levels of pain, and satisfaction with the services provided during the isolation period. We computed the descriptive statistics to explain the profile of participants, ANOVA was used to find out the difference between various groups on patient satisfaction, and structural equation modeling was performed to examine the link between pain levels, demographic factors, and patient satisfaction. This analysis was performed with STATA and SmartPLS-3.

**Results.** The findings of this study ascertained that respondent's education (Beta, $\beta = 2.961$, $p = 0.003$), assets such as possession of a house ($\beta = 4.563$, $p = 0.000$), perception of income adequacy during the lockdown ($= 2.299$, $p = 0.022$), and negligence in prevention ($\beta = 1.975$, $p = 0.049$) significantly affects patient satisfaction during the isolation period. Moreover, satisfaction with life, physicians and medicine, income adequacy, and fear of the participants was significantly linked to the pain level (Model F-stat = 86.323, $p = 0.0001$). Patients who were satisfied with their treatment had significantly lower odds of having pain (OR 0.023, 95% CI [0.001–0.0452]). Patients who had enough income to meet their basic necessities were found to have significantly lower odds of having pain (OR 0.155, 95% CI [0.032–0.740].

**Discussion and Conclusion.** We conclude that a higher level of education, low income, and living in a rented house contributes significantly to the feeling of more satisfaction with the provided services. This implies that people with more affluent backgrounds are less likely to be satisfied with the provided services as low-income and living in rented house groups are more satisfied during isolation than others. The pain level

is also affected by subjective factors such as fear and satisfaction which need to be considered while doing patient management. This study can be helpful in improving the delivery of public services of isolation centers by considering various characteristics and demographic factors of patients.

# INTRODUCTION

The COVID-19 pandemic has jolted the whole world. The outbreak started in China in late 2019 and turned into a pandemic due to its highly transmittable nature (*Lu, Liu & Jia, 2020*). In Pakistan, until October 18, 2020, 323,019 persons contracted the disease, whereas 6,654 persons died due to COVID-19 (*Ministry of National Institute of Health, 2020*). However, mRNA vaccines are fully developed to combat the severe consequences of COVID-19 by producing antibodies to combat the disease. The longevity of antibodies and their effectiveness against different variants of the virus remains a mystery. The most effective method to reduce the viral spread of highly transmittable diseases has been community-wide containment through quarantines and isolations (*Centre for Disease Control and Protection, 2020*). The practice of timely quarantine and isolation during previous outbreaks has shown results curtailing the spread of the disease (*Giubilini et al., 2018*).

It is often argued that quarantine and isolation are two sides of the same coin, but there are two folds of dissimilarity in terms of definition. Quarantine means isolating the persons who have been potentially exposed to the virus and are not sick yet to observe the virus's symptoms and reduce the potential spread of the disease. On the contrary, isolation is the separation of confirmed infected patients (*Brooks et al., 2020*). The incubation period of the virus is 1–14 days (*World Health Organization, 2020*). This created worry for the public about the isolation and management of isolation. Therefore, quarantine and isolation facilities are established across the world. In Pakistan, hundreds of places converted into temporary quarantine centers to facilitate quarantine and in hospitals, separate isolation wards were made for COVID-19 patients.

Quarantines and isolations of patients have played a vital role in controlling the spread of the disease. On the one hand, they facilitate in reducing the infection, but, on the other, cause many side effects, which is recently argued as shadow pandemic. The psychological side effects of isolation include anger, depression, anxiety, loss of patient satisfaction and mental wellbeing, and loss of self-control (*Abad, Fearday & Safdar, 2010*; *Rees et al., 2000*). Quarantine is also known to create a cardiovascular risk due to social isolation and depression (*Mattioli et al., 2020*). Due to the known effects of isolation, it is deemed important to investigate the socio-psychological effects of isolation on COVID-19 patients. Thus, the isolation experience can be improved so that the positive effects dominate the
negative side effects. These negative psychological effects also hinder the care and treatment of the patient by creating an overall negative impact on the patient's mind and body.

Developing countries are already struggling with the scarce health facilities and services to combat the disease. Where a patient in isolation experiences fear, anxiety, and depression (*Maunder et al., 2003*). Due to limited resources, a negative experience of the isolation wards can add to the discouragement of people to come to hospitals for recovery and may also cripple their belief in the health system of the country. Past studies provide evidence that pain management practices improve the health of a patient (including patients with chronic diseases) experience (*Soin et al., 2020*; *Anthony Jnr, 2021a*) and innovative methods such as digital care can be helpful during pandemic (*Anthony Jnr, 2021b*), but the COVID-19 may lead to aggravated perceived pain of individuals (*Song et al., 2020*). The negative experience is quite evident during isolation. Therefore, there is a need to investigate the factors that may help improve the isolation experience of patients and factors that, if controlled, can reduce the perceived pain of the patients of COVID-19. Additionally, the need is to investigate the experience of COVID-19 patients so that their experience can be improved in the future and pain can be mitigated. Factually, reduction in pain improves patient satisfaction (*Baker et al., 2007*; *Scher et al., 2018*). To address this research question, we focused on the experiences of the patients isolated in the COVID-19 ward. We hypothesized that patient's satisfaction during isolation might differ according to the distinct demographics of the patient. Besides, it may also be affected by different levels of pain and other factors, whereas fear and other factors may affect the pain levels (*Asmundson, Vlaeyen & Crombez, 2004*; *Keefe, Abernethy & Campbell, 2005*). Secondly, we focused on investigating the difference in patient satisfaction of COVID-19 patients with distinct demographics. Later, we investigated the effect of different factors on isolation, pain level, and patient satisfaction in separate models. The specific research questions are:

- Does patient's satisfaction during COVID-19 isolation differ among demographic groups?
- What are the factors affecting the pain level (subjective) of the COVID-19 patient during isolation?
- What are the factors affecting patient's satisfaction during COVID-19 isolation?

The introduction section is followed by the Materials and Methods section, which explains data collection, sample, and data analysis technique. The third section includes descriptive statistics, patient satisfaction among patients, factors affecting pain level, and factors affecting patient satisfaction. Results are followed by discussion and conclusion.

## MATERIALS & METHODS

This study relies on primary data collection for making inferences. We conducted a cross-sectional survey to collect the information from the COVID-19 patients who were isolated in the government centers. The study focused on the investigation of patient satisfaction and their pain perception of isolated patients. The variable of fear comprises of seven dimensions. The questionnaire adopted the fear scale from the study conducted by *Ahorsu et al. (2020)*. The pain level was measured using 5 point measurement scale based on

literature (*Huskisson, 1974*). Patient satisfaction is based on the scale based on Likert type scale. The questionnaire was developed based on literature (*Abad, Fearday & Safdar, 2010*; *Rees et al., 2000*; *Tang et al., 2020*) and was validated by experts, hospital management, and people working in isolation wards with COVID-19 patients. Patient satisfaction has been operationally defined in this study as the satisfaction of COVID-19 patients with the services provided in the public isolation center. Patient satisfaction is measured by asking questions about satisfaction with overall services, nurses, doctors, medicine, food, cleanliness, and behavior of peer patients. The Cronbach alpha for the patient satisfaction scale for the current study is excellent (i.e., 0.93), and the fear of covid scale is moderate (0.61). The benchmark of Cronbach alpha is according to the literature (*Taber, 2018*). Since we measure pain through one item, Cronbach alpha was not calculated for this item (*Bland & Altman, 1997*).

## Data collection

The sample was collected from COVID-19 patients who were isolated in COVID-19 isolation wards. The sample data was collected from the patients during their isolation using purposive sampling. Purposive sampling is a selective sampling where the population with certain characteristic is selected. Based on the objective of the study, only the COVID-19 patients were purposively selected for the study. However, at next stage, the COVID-19 patients to be included in the sample were selected randomly (for more details on purposive sampling, see *Tongco (2007)*. Due to the highly transmittable nature of the virus, it was very difficult to get permission to access isolated patients. The isolated patients did not have their digital gadgets as well. With the help of paramedic staff, the data were collected from patients during their isolation to avoid the forgetting bias in reporting their responses. The paramedic staff who helped in the collection of the data was first trained to collect the data. Due to reservations imposed by the nature of the disease, we could only access the patients of one hospital in Rawalpindi. Rawalpindi Institute of Urology and Transplantation hospital has one of the largest isolation centers in Pakistan with 50 designated beds for COVID-19 patients and is equipped with the necessary equipment, including ventilators. The hospital gave ethical clearance for conducting the research (Ethical approval Letter Ref No. 201). No one could go into isolation wards except for the staff. The staff working in isolation wards of the Rawalpindi Institute of Urology and Transplantation, Rawalpindi, helped us to get the questionnaires filled from the patients. The data were collected during June–August 2020 in Rawalpindi when COVID-19 was at its peak. Patients were briefed about the purpose of the research, confidentiality of the responses and right to withdraw at any time during the research. Patients participated in the research with their voluntary consent knowing the purpose of the research. A separate written consent was not taken; however, verbal permission was taken by asking their willingness to give the data and filling out the questionnaire showed their informed consent. The questionnaire stipulated the purpose of the research and assured anonymity. In addition, the questions, revealing their identity were not asked.

## Sample

The sample size for this study was calculated through G-power software by selecting parameters including the effect size, which was 0.15, error probability being 0.05, power as 0.95, and a total number of predictors were 5. The required calculated sample required was 107. However, a questionnaire booklet was distributed among 110 participants. Among them, six refused to participate, and four booklets were 90 percent incomplete, so their responses were discarded. The response rate was 90.9%. Therefore, the final analysis was done on 100 participant's responses.

## Data analysis technique

Two software, including STATA and SmartPLS-3, were used to test the hypotheses of this study. Smart PLS is software for variance-based structural equation modeling (SEM) using the partial least squares (PLS) path modeling method. The benefits of the PLS approach consist of measurement settings, theoretical settings, practical attention, and distributional attentions (*Falk & Miller, 1992*). The PLS approach relies on the data and is an exploratory methodology. For the present study, SmartPLS was beneficial due to its prediction-oriented approach and did not need large sample sizes (*Chin & Newsted, 1999*). Moreover, PLS is used to determine the latent variables. The authors stated that the PLS approach uses a weighted sum of indicators for creating component scores of latent variables. Normality was checked through skewness and kurtosis through Statistical Package for the Social Sciences (SPSS). Results showed that data were normally distributed for all the variables as the values of skewness and kurtosis were found to be in in the acceptable range (skeweness: $\pm 2$, kurtosis: $\pm 3$) between 0.005–2.203 and no multicollinearity among variables was observed. Furthermore, a one-way analysis of variation (ANOVA) was used to test for the differences in patient satisfaction across the categories of training levels and marital condition.

To analyze various factors affecting the level of pain, an ordered logistical regression was used. Due to the orderly nature of the responses of the pain variable, that technique was chosen.

The general model of ordered logistic regression is:

$$\text{Prob}\left(y_i = j\right) = \text{B}\left(z_{j+1} - X_i \emptyset\right) - \text{B}\left(z_j - X_i \emptyset\right)\dots\dots\dots\dots\dots\dots\dots\dots \quad (1)$$

Substituting the value of B

$$\text{Prob}(y_i = j) = \frac{\exp\left(z_{j+1} - X_i \emptyset\right)}{1 + \exp\left(z_{j+1} - X_i \emptyset\right)} - \frac{\exp\left(z_j - X_i \emptyset\right)}{1 + \exp\left(z_j - X_i \emptyset\right)}\dots\dots\dots \quad (2)$$

where "i" is the number of observations. "j" is 1,…,5 and represent the values for y. "X" is the vector of independent variables, "Ø" is the coefficient vector and $z_j$ are the cut points of the distribution. The independent variables include fear, age, satisfaction with medicine during isolation, sufficiency of income, satisfaction with life and satisfaction with the doctors.

Structural equation modeling was applied to see the relationship of factors with patient satisfaction.

For the analyzing patient satisfaction, structural equation modelling (SEM) was used. The underlying model of SEM is linear model. The general linear model in vector form is given below:

$$y_i = y_i \Omega + \mu$$

where "$y_i$" is a vector of outcome variable and "$\Omega$" is a matrix of parameters constrained according to the association of variables. The independent variables for the pain satisfaction model are age, number of dependants, household size, education, house ownership, marital status, income adequacy, negligence and pain level. PLS software was used because it provided reliable results even with a small sample size and is suitable for predicting. Following *Hair, Ringle & Sarstedt (2013)* criteria, the structural model has been assessed via bootstrapping procedure by looking at the values of beta, R-square, and corresponding *t*-values in the research model among latent constructs. In addition, Hair and colleagues (2013) have also suggested ways of determining effect size (f-square). The predictive power of the research model was determined by the R-square value of the dependent variable, while the trajectory coefficients assess the strength of the hypothetical relations.

## RESULTS

### Descriptive statistics

As illustrated in Table 1, the mean age of the sample is 38 years, ranging between 19 and 76 years. Most COVID-19 patients were homeowners and heads of families. Most of the respondents reported that they were happy before the eruption of the COVID-19 pandemic. In addition, most of the participants who remained in an isolation center were graduates ($n = 31$) or postdoctoral ($n = 23$) in full-time employment ($n = 62$).

### Patient satisfaction among patients

Patient satisfaction is the perception of medical facilities and paramedic care (*Lochman, 1983*). We have tested whether patient satisfaction is different among the different levels of education, marital status, and gender. To achieve the objective, ANOVA is used, and the results are shown in Table 2. The first hypothesis is not supported for this data, which implies that quarantine satisfaction is similar across all variable categories. Overall, the mean values indicate that non-married persons are more satisfied with the isolation facility than married, widowed or divorced persons, but the difference in patient satisfaction across categories is not significant. On the other hand, those who have done graduation reported high-level satisfaction compared to those who have primary, secondary, or college-level education, and the difference between the categories is not significant. Furthermore, there is no difference in mean score in levels of satisfaction between females and males, which is in agreement with the literature where patient satisfaction is found to not vary with demographics (*Karaca & Durna, 2019*). Hence, from these results, it is evident that patient satisfaction does not vary in people with different demographics, keeping other things constant, which raises the need to investigate other factors apart from medical care that may affect patient satisfaction and may eventually improve the experience of patient satisfaction the COVID-19 patients.

**Table 1  Demographics of the respondents.**

| Variables | Categories | Frequency |
|---|---|---|
| Age | Mean = 38 | 88 |
| | Standard deviation = 13.1 | |
| House possession | Own | 52 |
| | Rent | 40 |
| | Other | 8 |
| Head of family | Yes | 53 |
| | No | 47 |
| Feel happy before pandemic | Not very happy | 16 |
| | Not happy | 12 |
| | Neutral | 17 |
| | Happy | 32 |
| | Very happy | 23 |
| | Strongly disagree | 2 |
| Education | Primary | 12 |
| | Secondary | 4 |
| | High school | 14 |
| | College | 15 |
| | Graduate | 31 |
| | Post graduate | 21 |
| Employment status | Full time | 62 |
| | Part time | 5 |
| | Unemployed | 21 |
| | Full time student | 4 |
| | Retired | 8 |

In the next section, we will see the factors affecting the pain level, whereas pain and other factors also affect the patient satisfaction of individuals.

## Factors affecting pain level

The level of pain is a subjective phenomenon and therefore affected by subjective factors beyond objective reasons. We examined the effect of fear and other factors on the level of pain and attempted to model the subjective component of pain. The respondents were asked about the pain level on a five-point scale ranging from no pain to unspeakable. The results have been analyzed using ordered logistic regression and shown in Table 3. Based on literature, the variables were identified. The significance of the variables is analyzed through $t$-test and the model through F-test. The robustness of the model is checked through adding and dropping of variables and the final model was selected based on Akaike Information Criteria (AIC) and Bayesian Information Criteria (BIC). The AIC and BIC criteria suggests Model 1 as the final model and is significant which is tested through F-test. According to the findings, when the respondent is neutral with respect to fear rather than disagreeing, there is a higher probability of having a higher level of pain. This suggests that patients being neutral about fear raises the likelihood of having a higher level of pain. In addition, as satisfaction with the medication supplied in the isolation facility increases, the

**Table 2 Analysis of variance estimating the difference in patient satisfaction between categories of demographic variables.**

|  | N | Mean (SD) | F | P-value |
|---|---|---|---|---|
| **Marital status** |  |  | 1.379 | 0.254 |
| 1. Unmarried | 27 | 2.872(0.931) |  |  |
| 2. Married | 68 | 1.500 (1.211) |  |  |
| 3. Widow | 2 | 2.071 (0.707) |  |  |
| 4. Divorced | 2 | 2.857 (1.111) |  |  |
| **Education level** |  |  | 1.673 | 0.149 |
| 1. Primary | 12 | 2.214 (1.336) |  |  |
| 2. Secondary | 4 | 2.320 (0.513) |  |  |
| 3. High school | 15 | 2.666 (1.072) |  |  |
| 4. College | 15 | 3.019 (1.292) |  |  |
| 5. Graduate | 27 | 3.218 (1.084) |  |  |
| 6. Postgraduate | 22 | 2.871 (1.020) |  |  |
| **Gender** |  |  | 0.119 | 0.730 |
| 1. Male | 48 | 2.816 (1.023) |  |  |
| 2. Female | 51 | 2.896 (1.125) |  |  |

probability of experiencing pain decreases. This indicates that if a person is satisfied with the treatment, then the sensing of pain will also be less as the patent will be anticipating to get better before long. The probability of having higher pain reduces if a person can afford some necessities compared to the situation where the income is not adequate. Income adequacy suggests the affordability of treatment and better, treatment may reduce pain and pain perception. An interesting result came out of the data, suggesting that patients with COVID-19 who are more satisfied with their lives are more likely to have more pain. People having contended life are more likely to report a higher level of pain subjectively. COVID-19 is known to be lethal. People with higher life satisfaction might fear the loss of life, and thus these emotions might interpret as higher pain. If the COVID-19 patients are neutral about satisfaction with the doctors as compared to disagreeing, the probability of perceiving higher pain is lower.

## Factors affecting patient satisfaction

Based on the literature, the pain level, and other demographic variable factors, significantly affect patient satisfaction. Results indicated that the model fits the data (SRMR = 0.049, Chi square = 182.36, NFI = 0.812). Bootstrapping is used to test this hypothesis. Results in Table 4 show that age, education level, house possession, income adequacy during the lockdown, negligence in preventing from corona, and levels of pain have a significant direct effect on patient satisfaction in public isolation centers.

The results indicate that patient satisfaction decreases with the increase in age. It was found that people with higher education were more satisfied with the isolation facilities. With the increase in the level of education, awareness can increase, and so the realization of the efforts of doctors and staff and the facilities provided make people more satisfied. Furthermore, people possessing their own home have lower satisfaction from isolation

**Table 3  Ordered logistic regression showing effect of different factors affecting pain level.**

| | Odds ratio | |
|---|---|---|
| VARIABLES | Model 1 | Model 2 |
| Fear | | |
| Strongly agree | 12.64 | 2.686 |
| | (0.389–410.1) | (0.115–62.79) |
| Agree | 10.25 | 5.870 |
| | (0.430–244.3) | (0.327–105.3) |
| Neutral | 39.55[**] | 66.72[***] |
| | (1.792–872.7) | (3.196–1,393) |
| Disagree | 0.186 | 0.387 |
| | (0.00869–3.975) | (0.0213–7.009) |
| Base category: strongly disagree | | |
| Age | 0.940[**] | 0.963[**] |
| | (0.889–0.994) | (0.915–1.014) |
| Satisfaction with medicine during isolation | | |
| Dissatisfied | 0.01000[**] | 0.131[*] |
| | (0.000108–0.929) | (0.0150–1.149) |
| Neither satisfied nor dissatisfied | 0.00389[**] | 0.0476[**] |
| | (4.47e−05–0.338) | (0.00386–0.587) |
| Satisfied | 0.0230[**] | 0.0734[**] |
| | (0.00117–0.452) | (0.00856–0.630) |
| Very satisfied | 0.0541[**] | 0.193 |
| | (0.00427–0.686) | (0.0250–1.494) |
| Base category: Very dissatisfied | | |
| Sufficiency of income | | |
| Can meet necessities | 0.155[**] | 0.376[**] |
| | (0.0324–0.740) | (0.0876–1.616) |
| Can afford somethings | 0.499 | 0.370 |
| | (0.109–2.283) | (0.0885–1.545) |
| Can afford everything | 5.131 | 1.645 |
| | (0.547–48.13) | (0.292–9.276) |
| Afford everything and still save money | 2.040 | 4.389e+11 |
| | (0.01 –0.02) | (0.01 -0.02 ) |
| Base category: Not at all adequate | | |
| Satisfaction with life | | |
| Disagree | 122.0[***] | 39.77[***] |
| | (6.887–2,162) | (2.846–555.7) |
| Neutral | 744.9[***] | 47.88[***] |
| | (19.59–28,329) | (3.103–738.7) |
| Agree | 61.06[**] | 8.329 |
| | (2.556–1,458) | (0.614–113.0) |

**Table 3** (*continued*)

|  | Odds ratio | |
| --- | --- | --- |
| **VARIABLES** | **Model 1** | **Model 2** |
| Strongly agree | 12.63 | 5.612 |
|  | (0.0634–2,514) | (0.0641–491.4) |
| Base category: Strongly disagree |  |  |
| Satisfaction with the doctors |  |  |
| Disagree | 8.997 |  |
|  | (0.233–348.1) |  |
| Neutral | 0.0315[*] |  |
|  | (0.000656–1.515) |  |
| Agree | 0.806 |  |
|  | (0.0933–6.967) |  |
| Strongly agree | 0.97 |  |
|  | (0.1–1) |  |
| Base category: Strongly disagree |  |  |
| Constant cut1 | 0.141 |  |
|  | (0.00226–8.758) |  |
| Constant cut2 | 2.515 |  |
|  | (0.0458–138.2) |  |
| Constant cut3 | 24.55 |  |
|  | (0.420–1,435) |  |
| Constant cut4 | 320.1[***] |  |
|  | (4.730–21,668) |  |
| Pseudo R square | 0.323 | 0.269 |
| F statistics | 86.323 | 71.90 |
| F statistics *p*-value | 0.000 | 0.000 |
| AIC | 230.279 | 238.693 |
| BIC | 292.213 | 293.194 |
| Observations | 88 | 88 |

**Notes.**

Dependent variable: Pain level.

[*]*p*-value < 0.1.
[**]*p*-value < 0.05.
[***]*p*-value < 0.01.

centers than people living in a rental home. People feel better about isolating themselves at home. In addition to this, the results illustrated that those individuals who had enough income to meet their expenses during lockdown reported a low level of satisfaction while having inadequate money increase isolation centers' satisfaction. Those who perceived that they were suffering from COVID-19 due to their negligence displayed a high level of satisfaction. Neglect has been operationally defined as the carelessness in taking steps to protect themselves from COVID-19. People whose income is adequate are less satisfied with patients. Similarly, having a high level of pain enhances the level of satisfaction at the isolation center. They feel satisfied when they get medical facilities and other services in public isolation centers, which otherwise was not possible at home while facing a high

**Table 4** Summary of Construct's R square, Beta Coefficient, *t*-value and Significance level.

| Hypothesis | Path/Beta Coefficient | *t*-value | *p*-value |
|---|---|---|---|
| Age | −0.222 | 2.231 | 0.026[**] |
| Number of dependents | 0.024 | 0.236 | 0.813 |
| Education | 0.276 | 2.961 | 0.003[***] |
| Household size | 0.097 | 0.820 | 0.413 |
| House ownership (own house) Base category: rented house | −0.404 | 4.563 | 0.000[***] |
| Income adequacy | −0.237 | 2.299 | 0.022[**] |
| Marital Status (married) (Base category: unmarried) | −0.084 | 1.005 | 0.315 |
| Negligence | 0.201 | 1.975 | 0.049[**] |
| Pain Level | 0.348 | 3.420 | 0.001[***] |

**Notes.**

Dependent variable: Patient satisfaction.

[*]*p*-value < 0.1.

[**]*p*-value < 0.05.

[***]*p*-value < 0.01.

level of pain. On the other hand, several dependents, household size, and marital status are not significantly related to the satisfaction of isolation centers. As a result, the second assumption is partially supported. Table 4 shows the t-values, standardized path/ beta coefficients, and outcomes. Overall, the bootstrap analysis indicated that demographic variables added to the model explained 30.4% of the variance in quarantine satisfaction.

## DISCUSSION

The COVID-19 pandemic has jolted the whole world with infectious disease by spreading the sickness and causing mortalities. According to WHO guidance, quarantine and isolation are among the primary methods to reduce the spread of the disease. The literature has shown that isolation may have negative effects on the patients (*Mattioli et al., 2020*), so it is very important to improve patients' experience by providing them a better environment to recover quickly.

Pain is a subjective experience (*Koyama et al., 2005*; *Raja et al., 2020*). The subjective factors affecting pain are fear, satisfaction with life, income adequacy, satisfaction with doctors and medicine. The age also affects the pain level, but the effect size is very low as the odds are close to 1. The literature says that pain might be due to fear (*Crombez et al., 2013*; *Markfelder & Pauli, 2020*). Fear of pain aggravates the feeling of pain (*Crombez et al., 2013*), and this study found the same on COVID-19 patients. The satisfaction with the doctors and medicine decreases the perception of pain as the person trusts the quality of care given. Similarly, income adequacy brings the affordability of the treatment and less worry about the finances, and hence, perceived income adequacy affects the pain (*Crombez et al., 2013*). Compared to those who strongly disagree, if a person demonstrates that they are satisfied with their life, the likelihood of experiencing greater pain is high. The individual's satisfaction corresponds to his overall cognitive assessment of life (*Bergefurt et al., 2019*). Literature reported that more social people tend to report higher satisfaction, happiness

(*McCarthy & Habib, 2018*). Loneliness not only affects life satisfaction, but also affects health (*Bergefurt et al., 2019*; *Hwang et al., 2020*). The isolated patients having higher life satisfaction face amplified effects of isolation in terms of emotional and health, wellbeing, which is translated into reporting a higher level of pain.

*Sprang & Silman (2013)* found that individuals in isolation had more post-traumatic disorders than those who were not in isolation. Based on the primary data collected from the COVID-19 patients, we found that their satisfaction was not significantly different among people with varying education, gender, and marital status. Outcomes are consistent with patient satisfaction literature. An estimated 40% of people are satisfied with the isolation facility. Patient satisfaction is affected by income adequacy, house ownership, age, education, negligence, and pain level. Age has a direct effect on patient satisfaction. Patient satisfaction was low among younger patients, as it was not frequent that they were ill or hospitalized. People with adequate income tend to have better health and expect to get high-quality care, and in case there is a discrepancy in the car, they get more dissatisfied than those who do not have a higher financial status. People who do not have adequate income are already negligent of health and health care and do not perceive higher dissatisfaction (*Karaca & Durna, 2019*). Education gives awareness and the ability to cope with the disease and positively relates to patient overall satisfaction (*Dzomeku et al., 2013*; *Wudu, 2021*). Negligence in taking preventive measures for COVID-19 has an important effect on patient satisfaction. Highly negligent people taking preventive steps to protect themselves from COVID-19 have higher patient satisfaction as the medical facilities and services make them feel in a safe environment that they were missed in the first place. Patient satisfaction is extremely important for hospital patient satisfaction literature proves that people with a lower perception of quality care tend to complain more about the hospital (*Stelfox, Bates & Redelmeier, 2004*).

This study has far-reaching implications for the health care system and patients. The healthcare system needs to improve their isolation facilities though struggling with scarce resources. It will help them to reduce the pain level of patients and improve their experience. With closed borders and a ban on foreign travel, local treatment is the only option for COVID-19 patients. The improved isolation experience will help patients to spread good word of mouth, which will help regain the nation's trust in doctors, paramedic staff, and the health care system despite having scarce resources. This will also help them ensure that the best is being provided to them. They will eventually enjoy the isolation experience, which will not help them recover, but their safe isolation will also protect others from the spread of the disease. Other countries can also derive lessons from this study by focusing on the factors identified to improve isolated patients' experience in isolation centers.

The data collection from isolated COVID-19 patients during isolation was a big hurdle. We did not want to ask patients about their experience after isolation to avoid the bias of forgetting affecting our results. Due to the stated hurdle of access to patients isolated in centers, we could not collect large data. Further research can be done with a large data sample and in different countries to investigate individuals' isolation experience and see the country-specific factors.

## CONCLUSIONS

The novelty of COVID-19 has not only brought issues in the disease treatment and management, but also the overall medical care and related services. We cannot ignore the importance of treatment, recovery and prevention in this pandemic. The patient experience is extremely important in isolation centers to build patient confidence in paramedics and physicians. A better experience will also contribute to reducing pain for patients. The factors affecting the pain level of patients are satisfaction with doctors and medicine, satisfaction with life, income adequacy, and fear. An improved isolation facility with quality of doctors and medicine can help to reduce the pain level of the patients.

It is important to maintain isolation facilities for patients by providing a comfortable environment to patients where they can recover and feel satisfied and perceive less pain. Improving the patient experience may improve the overall quality of hospitals and, as a result, hospitals will face fewer complaints. Being a developing country, income adequacy is also an important factor affecting pain and patient satisfaction. Satisfaction with medical care helps the individual to perceive less pain.

Creating public isolation facilities and equipping them with the latest available facilities is a great initiative, especially in developing countries where masses cannot afford these facilities at an out-of-pocket expense. The focus on the provision of services can improve their experience and thus patient satisfaction. However, patient satisfaction is not significantly different between the distinct groups of the population. Improving overall facilities may help to increase the patient satisfaction of patients. Reduction of fear through proper counseling and media awareness may help to bring down the perceived pain of patients. Improving isolation experience will also help decrease negligence in other patients about their diseases and approach the hospitals for isolation, believing that they are in trusted hands.

### Funding
The authors received no funding for this work.

### Competing Interests
The authors declare there are no competing interests.

### Author Contributions
- Wajiha Haq and Siddrah Irfan conceived and designed the experiments, performed the experiments, analyzed the data, prepared figures and/or tables, authored or reviewed drafts of the paper, and approved the final draft.
- Syed Hassan Raza performed the experiments, analyzed the data, prepared figures and/or tables, authored or reviewed drafts of the paper, and approved the final draft.
- Waqar Ahmed, Samia Wasif and Saeed Ahmad performed the experiments, prepared figures and/or tables, authored or reviewed drafts of the paper, and approved the final draft.

- Mian Alam performed the experiments, prepared figures and/or tables, and approved the final draft.
- Muhammad Wasif Malik conceived and designed the experiments, performed the experiments, prepared figures and/or tables, authored or reviewed drafts of the paper, and approved the final draft.

## Human Ethics

The following information was supplied relating to ethical approvals (i.e., approving body and any reference numbers):

Institute of Urology and Transplantation, Rawalpindi, Pakistan approved this research (Ethical approval Letter Ref No. 201).

## Data Availability

The raw data are available as a Supplemental File.

## Supplemental Information

Supplemental information for this article can be found online at http://dx.doi.org/10.7717/peerj.11859#supplemental-information.

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
