# Peer review of "Pain and satisfaction: the case of isolated COVID-19 patients of Pakistan"

_PeerJ, doi:10.7717/peerj.11859_

## Round 0.1 · original submission · Major Revisions

There are some issues that have been highlighted by the reviewers in their reports. Please, try to apply the suggested changes in a revised version of the text.

·

Basic reporting

- The paper needs proofreading and grammatical editing as grammar issues have been prevalent throughout the manuscript. I would recommend rewriting various sentences and subject/verb agreement issues are widely present all over the text.
- As for the article, it is well structured in general.
- The abstract needs to include numbers.
- Old references in the discussion need to be updated.
- Add an implications section

Experimental design

- The research design is not well described.
- Chronbach's alpha for the questionnaire was not assessed or mentioned.
- Statistical analysis was undertaken as appropriate.
- Add response rate
- Add the number of ethical approval
- Add sampling methodology and related reference

Validity of the findings

- Add frequencies to table 1
- Results are adequate. However, more tests are advised to enrich the findings and giving a higher impact to the paper.

Reviewer 2 ·

Basic reporting

The study is timely and deserves merit.

Experimental design

The methodology employed is valid but few clarifications are required. (see comments below).

Also, do include a model that shows the hypotheses tested.

Validity of the findings

Findings are well presented.

Additional comments

Pain and satisfaction: the case of isolated COVID-19 patients of Pakistan
Below are comments to improve the manuscript
Include tool used for analysis in the abstract (STATA and Smart PLS-3).
Elaborate more on the issues and need for the research in the introduction section.
Also, provide research questions to be examined in the study in the introduction section.
Add a paragraph of the summary of the sections in the last sentence in the introduction section.
Include literature review section (related works) section as section 2, that reviews prior COVID-19 studies such as
• Amol Soin, M. D., Srinivas Vuppala, M. D., Gregory Surfield, M. D., Ricardo Buenaventura, M. D., Mark Malinowski, D. O., Rajaratnam, A., ... & Chandoke, A. (2020). Ohio response to COVID-19 and its impact on interventional pain management practices. Pain physician, 23, S439-S447.
• Anthony Jnr, B. (2021). Implications of telehealth and digital care solutions during COVID-19 pandemic: a qualitative literature review. Informatics for Health and Social Care, 46(1), 68-83.
• Song, X. J., Xiong, D. L., Wang, Z. Y., Yang, D., Zhou, L., & Li, R. C. (2020). Pain management during the COVID-19 pandemic in China: Lessons learned. Pain Medicine, 21(7), 1319-1323.
• Anthony Jnr, B. (2021). Integrating telemedicine to support digital health care for the management of COVID-19 pandemic. International Journal of Healthcare Management, 1-10.
Also, do include a model that shows the hypotheses tested.
The methodology is well written but provide more information on the sampling technique employed.
Provide full meaning of STATA and Smart PLS-3 and why was STATA and Smart PLS-3 employed and not SPSS?
There are results on factors affecting pain level. How was the factors derived and do present these factors in a less vague approach as this is one of the contribution of the study?
Include standard deviation for descriptive analysis.
How was reliability and validity measured?
Include a section on implications of the study.
Provide discussion on the limitations and future works in the conclusion section.

---

## Round 0.2 · Minor Revisions

1. Include numbers in the abstract (results).
2. Give more details about “The sample data was collected from the patients during their isolation using purposive sampling (for purposive sampling see Tongco (2007)).”
3. Include a subsection in Discussion so as to address the Implications to research and clinical practice, and another for the study limitations.

---

## Round 0.3 · Minor Revisions

Implications and limitations should be placed before the conclusions.

Reviewer 2 ·

Basic reporting

Well presented as the manusrript was revised.

Experimental design

Acceptable

Validity of the findings

Can be improved as some results are missing.

Additional comments

Pain and satisfaction: the case of isolated COVID-19 patients of Pakistan
Below are comments to improve the revised version of the manuscript

Also, provide research questions to be examined in the study in the introduction section. The research question should be in bullet point

Include literature review section (related works) section as section 2, that reviews prior COVID-19 studies such as
• Amol Soin, M. D., Srinivas Vuppala, M. D., Gregory Surfield, M. D., Ricardo Buenaventura, M. D., Mark Malinowski, D. O., Rajaratnam, A., ... & Chandoke, A. (2020). Ohio response to COVID-19 and its impact on interventional pain management practices. Pain physician, 23, S439-S447.
• Anthony Jnr, B. (2021). Implications of telehealth and digital care solutions during COVID-19 pandemic: a qualitative literature review. Informatics for Health and Social Care, 46(1), 68-83.
• Song, X. J., Xiong, D. L., Wang, Z. Y., Yang, D., Zhou, L., & Li, R. C. (2020). Pain management during the COVID-19 pandemic in China: Lessons learned. Pain Medicine, 21(7), 1319-1323.
• Anthony Jnr, B. (2021). Integrating telemedicine to support digital health care for the management of COVID-19 pandemic. International Journal of Healthcare Management, 1-10.
Also, do include a model that shows the hypotheses tested.
How was Normality tested in the study if SPSS was not used.
Provide full meaning of SPSS
There are results on factors affecting pain level. How was the factors derived and do present these factors in a less vague approach as this is one of the contribution of the study?
Include standard deviation for descriptive analysis.
How was reliability and validity measured?
The section implications of the study should be placed before the conclusion section

---

## Round 0.4 · accepted · Accept

All the reviewers' concerns have been correctly addressed.

·

Basic reporting

The author has incorporated the needed changes according to comments.

Experimental design

The author has incorporated the needed changes according to comments.

Validity of the findings

The author has incorporated the needed changes according to comments.

Additional comments

Dear author,
Thank you for your revision.

Reviewer 2 ·

Basic reporting

The paper is grounded on the literature. Also, the research questions employed has been provided.

Experimental design

The methodolofy employed is well presented. Besides, the sampling method has been presented.

Validity of the findings

Data analysis was performed with STATA and SmartPLS-3. Results are valid.

Additional comments

The manuscript was revised as recommended.